# Spurious Features in Continual Learning

## Abstract

Continual Learning (CL) is a field of research that addresses training scenarios where the data distribution changes over time. One of its key challenges is learning without forgetting. To achieve this, CL algorithms need to learn stable and robust representations that can generalize to new data. However, since data is acquired gradually, the learned representations may be biased and require validation with more data. This paper investigates the impact of spurious features on CL algorithms. We show that these algorithms can learn to rely on features that are not generalizable, leading to poor performance in both memorization and generalization. We identify two related problems: (1) spurious features (SF), which result from a shift in the data distribution between training and testing, and (2) local spurious features (LSF), which arise due to the limited access to data at each training step. To study the impact of (1), we conduct a series of experiments that vary the amount of spurious correlation of the data distribution. We also propose an experimental setup to estimate the influence of (2) in usual continual learning scenarios. Our results show that (1) and (2) can lead to model overfitting and lead to performance degradation in CL, in addition to catastrophic forgetting (CF). By highlighting the influence of (local) spurious features in CL algorithms, this paper offers a novel perspective on performance decrease in continual learning.

## 1 Introduction

Feature selection is a standard machine learning problem. Its objective is to improve the prediction performance, provide faster and more effective predictors, and provide a better understanding of the underlying process that generated the data (Guyon & Elisseeff, 2003). In this paper, we are interested in improving prediction performance in the presence of spurious features. Spurious features arise when the features' presence is predictive of labels in training data but not in test data. Learning algorithms that rely on spurious features will generalize badly to test data.

In continual learning (CL), the training data distribution changes over time. Hence, we could expect that spurious features (SFs) in one time step of the data distribution will not last. A continual learning algorithm relying on a spurious feature to solve a task can then be resilient and learn better features later, given more data. Algorithms can aim to detect and ignore spurious features learned in the past (Javed et al., 2020). An example of a task with spurious features could be a classification task between red cars and white bikes, but in test data, both are in a unique blue. A model could easily overfit the colour to solve the task while it is not discriminative in the test data. A covariate shift between train and test data notably causes this problem.

On the other hand, in CL, the second type of spurious feature can be described: *local spurious features*. Local features denote features are predictive of labels within a task (a state of the data distribution) but not in the full scenario. In opposition to the usual *spurious features*, this problem is provoked by the unavailability of all data at once, for example, we can assume that during its lifetime, the algorithm has seen bikes and cars of all colours. However, currently, only red cars and white bikes are available in train data. The algorithm can overfit the colour feature because it provides a trivial solution at the moment. Even if the test data is representative of the overall data distribution and there is no significant covariate shift between all train data and test data, the features learned locally lead to poor generalization.

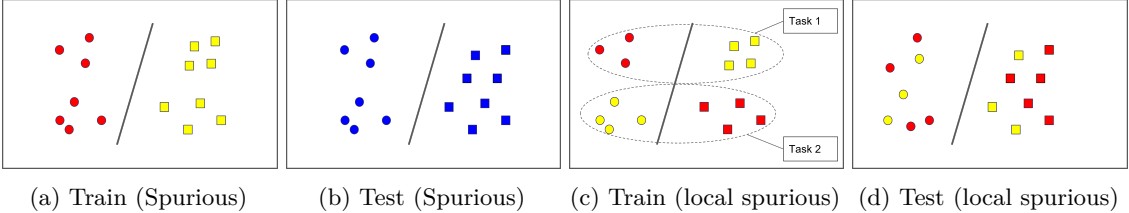

(a) Train (Spurious)  (b) Test (Spurious)  (c) Train (local spurious)  (d) Test (local spurious)

Figure 1: **Spurious features and local spurious features.** If the task is to distinguish the squares from the circles. In Fig. 1a and 1b, the colour is a spurious feature because there is a covariate shift between train and test data. In Fig. 1c and 1d, we observe two tasks of a domain-incremental scenario, the colours are locally spurious in tasks 1 and 2. Even if there is no significant covariate shift between train and test full data distribution, colours appear discriminative while looking at data within a task.

This paper investigates both the problem of spurious features (with covariate shift) and local spurious features (without covariate shift) in CL as shown in Fig. 1. Our contributions are: (1) We propose a methodology to highlight the problems of spurious features and local spurious features in continual learning. (2) We create a binary CIFAR10 scenario *SpuriousCIFAR2* inspired by coloured MNIST to experiment with spurious correlations. (3) We propose a modified version of Out-of-Distribution (OOD) generalization methods for continual learning and evaluate them on *SpuriousCIFAR2*. (4) We identify local spurious features as a core challenge for continual learning algorithms along with catastrophic forgetting.

We expect this paper to improve the understanding of the continual learning problem.

## 2    Related Work

In large part of the continual learning bibliography, algorithms assume that to avoid catastrophic forgetting (CF), they should not increase the loss on past tasks (Kirkpatrick et al., 2017; Ritter et al., 2018). It leads to the definition of interference/forgetting of (Riemer et al., 2019): $\frac{\partial L(x_i, y_i)}{\partial \theta} \cdot \frac{\partial L(x_j, y_j)}{\partial \theta} < 0$, $\forall (x_i, y_i) \in \mathcal{T}_i$ and $\forall (x_j, y_j) \in \mathcal{T}_j$ with $j > i$, $< \cdot >$ is the dot product operator. Following this definition, increasing the loss on past tasks necessarily leads to a performance decrease. However, it does not take into account that the algorithm might have learned spurious/local features that need to be forgotten to improve overall performance. The loss might have to be temporarily increased to reach a more general solution, and optimizing the interference equation could be counterproductive. On the same line, the presence of spurious features could be adversarial to most continual regularization strategies. For example, if we measure weights' importance with Fisher information, high importance will be given to weights using spurious features, and regularization will penalize their modification. Regularization could protect features that generalize poorly to the test set.

Vanilla rehearsal or generative replay is a good solution to not forget meaningful information and to deal with spurious features. By replaying old data, algorithms simulate an independent and identical distribution (iid) and avoid local spurious feature problems. Replay methods have been shown in the bibliography to be efficient and versatile even in their most straightforward form (Prabhu et al., 2020). Notably, CL state-of-the-art on ImageNet uses replay (Douillard et al., 2020; Zhao et al., 2020).

The research field that usually deals with spurious correlations is the out-of-distribution (OOD) generalization field. This field has received a lot of attention in recent years, especially since the Invariant Risk Minimization (IRM) (Arjovsky et al., 2019) paper. OOD approaches target training scenarios where there are several training environments within which different spurious features are predictive of labels. The goal then is to learn invariant features among all environments to build an invariant predictor in all training environments and potentially any other (Arjovsky et al., 2019; Ahuja et al., 2021; Sagawa et al., 2019; Pezeshki et al., 2020). This paper will adapt some of those approaches for continual learning to evaluate how those approaches can deal with sequences of tasks.

# 3 Problem Formulation

This section introduces the spurious feature problems in a sequence of tasks. The goal is to present the key types of features, namely: general, local, and spurious features.

## 3.1 General Formalism

We consider a continual scenario of classification tasks. We study a function $f_\theta(\cdot)$, implemented as a neural network, parameterized by a vector of parameters $\theta \in \mathbb{R}^p$ (where p is is the number of parameters) representing the set of weight matrices and bias vectors of a deep network. In continual learning, the goal is to find a solution $\theta^*$ by minimizing a loss $L$ on a stream of data formalized as a sequence of tasks $[\mathcal{T}_0, \mathcal{T}_1, ..., \mathcal{T}_{T-1}]$, such that $\forall (x_t, y_t) \sim \mathcal{T}_t$ $(t \in [0, T-1])$, $f_{\theta^*}(x) = y$. We do not use the task index for inferences (i.e. single head setting).

To describe the different types of features, let $z$ be a feature and $x \sim \mathcal{D}$ a datum point in dataset $\mathcal{D}$. We define $w(.)$ as a function which returns 1 if $z$ is in $x$

Table 1: Summary of characteristics of the types of features. For a feature $z$ of a class $c$, ✓ denote if a feature is discriminative in a given set of data, i.e. it verifies eq. (1). The set of data are: a single task $\mathcal{T}_t$, the whole scenario $\mathcal{C}_T$, the test set $\mathcal{D}_{te}$.

| Name | $\mathcal{T}_t$ | $\mathcal{C}_T$ | $\mathcal{D}_{te}$ |
|---|---|---|---|
| Good/Global Feature ($z_+$) | ✓ | ✓ | ✓ |
| Spurious Feature ($z_{spur}$) | ✓ | ✓ | × |
| Local Feature ($z_{loc}$) | ✓ | ? | ? |
| Local Spurious Feature ($z_{spur:t}$) | ✓ | × | × |

and 0 if not. $w(.)$'s output is binary for simplicity. Then, for all data with a label $y$ in the dataset $\mathcal{D}$, we can estimate the relationship between the presence of a feature and the label $r(\mathcal{D}, z, y)$ as the probability $P(Y = y | w(z, x) = 1; x \sim \mathcal{D})$, which estimates how a feature can be used for prediction. We can then define discriminative features as:

$z$ is discriminative for class $y$ in $\mathcal{D}$ if:

$$1 - r(\mathcal{D}, z, y) < \epsilon \tag{1}$$

$\mathcal{Y}$ is the set of classes in $\mathcal{D}$, $\epsilon$ is a small real value. In other words, $z$ is discriminative for $y$ if its presence is statistically predictive of the label. We can note that $z$ can be discriminative even if it is predictive only for a subset of the class samples. Then a good feature $z_+$ for a class $y$ respects (1) for training data $\mathcal{D}_{tr}$ and test data $\mathcal{D}_{te}$.

## 3.2 Spurious Features (SF) and Local Spurious Features (LSF)

A spurious feature $z_{spur}$ for a class $y$ respects (1) for training data $\mathcal{D}_{tr}$ but not for test data $\mathcal{D}_{te}$. A spurious feature is predictive of the labels in training data but not in testing data. Hence, learning from $z_{spur}$ may offer a low training error but a high test error. The presence of $z_{spur}$ is due to a covariate shift between train and test distribution which changes the feature distribution.

In continual learning, the covariate shift between train and test $z_{spur}$ may also lead to poor generalization. Further, the features can be locally spurious, e.g., they are predictive of labels within a task but not within the whole scenario. We name them *local spurious features* (LSF). We illustrate the difference between spurious features and local spurious features in Figure 1.

At task $t$, A local spurious feature $z_{spur;t}$ respects (1) for a class $y_t$ in task $\mathcal{T}_t$, but not for the whole scenario $\mathcal{C}_T$. $z$ is a LSF for a class $y$ in $\mathcal{T}_t \sim \mathcal{C}_T$, with $t \in [\![0, T-1]\!]$:

$$1 - r(\mathcal{T}_t, z, y) < \epsilon$$
$$\text{and } 1 - r(\mathcal{C}_T, z, y) > \epsilon \tag{2}$$

$\mathcal{Y}_t$ is the classes set in task $\mathcal{T}_t$ and $\mathcal{Y}$ is the classes set in the full scenario $\mathcal{C}_T$ composed of $T$ tasks.

A LSF $z_{spur;t}$ is predictive of a label on the current task but not on the whole scenario. $z_{spur;t}$ can be extended from a single task $\mathcal{T}_t$ to all task seen so far $\mathcal{T}_{0:t}$ without loss of generality.

### 3.3 Global vs Local Solution:

Machine learning models solve tasks by relying on features that are statistically predictive of labels. Then, while learning on a task $t$, we can distinguish a local solution $\theta_t^*$, satisfying for the current task $\mathcal{T}_t$, from a global solution $\theta_{0:T}^*$ that is satisfying for whole scenario $\mathcal{C}_T$ (past, current, and future tasks).

Similarly, we can differentiate local and global features, contributing to local and global solutions. A global feature (or good feature) is a feature $z_+$ that is always discriminative (c.f. table 1). Unfortunately, at time $t$, we can not know if a feature is global without access to the future. Therefore, algorithms should learn with their current data but update their knowledge afterwards, given new data. For example, in classification, the discriminative features for a given class depend on all the classes. Therefore, when new classes are added to a classification task (as is the case in class-incremental scenarios), features previously discriminative can become not discriminative.

### 3.4 Spurious features: Different cases

In a given scenario, if we know a priori the characteristics of spurious features, we can potentially address them. Hence, in this section, we identify different cases among the spurious features which could lead to different assumptions useful to reduce their influence. Those different cases can either apply to SFs or LSFs.

**Invariance:** *(1) Fully observable features* (Javed et al., 2020) The global features are always observable. In this case, we can assume that features not invariant are spurious, and we can learn to ignore them. *(2) Partially observable features* The global features' presence is not invariant. In this case, features that do not last can either be spurious or good.

**Noisiness of Spurious Features:** *(1) SFs are irrelevant for other classes* they can be ignored completely without affecting the learning process. We will refer to them as noisy SFs. *(2) SFs are good features for other classes*, e.g., for classification, the colour can be a spurious feature for some classes and valuable for others. We can not ignore those features since it could lead to poor performance in other classes.

In our experiment, we propose settings with fully observable in Sec. 4 and 5 and partially observable data in Sec. B. Our spurious features are noise in Sec. 4 and B and either noise or true features in Sec. 5. Nevertheless, we approach the settings without assumption and without exploiting exploiting information about spurious feature types.

## 4 Influence of Spurious Features (SFs) on Continual Learning

This experimental section studies how continual learning algorithms can deal with spurious features. We design a scenario with spurious features that change at each task. We create a set of scenarios with gradual correlations between spurious features and labels. We evaluate various baselines to assess continual learning capabilities (Sec. 4.3.2) in such scenarios. We also experiment with potential solutions to deal with spurious correlation.

### 4.1 Setting

**Setting Goal:** We want to create a setting to highlight how spurious features can disrupt CL algorithms and discuss the problems it can bring to existing approaches. Moreover, the setting should evaluates the capacity of algorithms to deal with spurious features.

**Data:** Inspired by coloured-MNIST binary setup Arjovsky et al. (2019), we propose a benchmark "SpuriousCIFAR2" with CIFAR10. We convert the ten-way classification dataset into a binary dataset (without label noise). The new classes are "transportation means versus not transportation means", i.e., cars, trucks, ships, airplanes, and horses versus the other classes: birds, cats, dogs, deers, and frogs. The goal is to have a simple classification setting, more challenging than coloured MNIST. We add to the original data spurious features. They are $2 \times 2$ pixels square of colour randomly positioned in the image. For each class, we sample a random colour for the squares (Figure 3). By default, all images have a square of colour, but We can vary

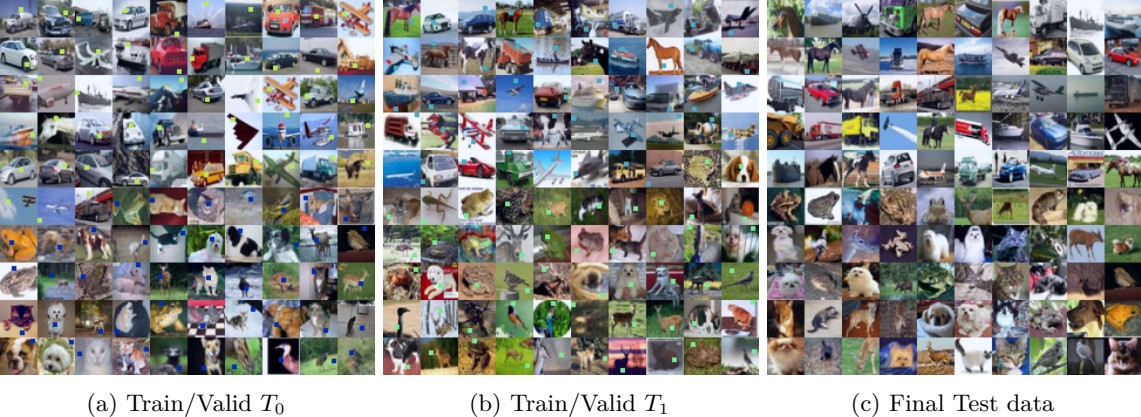

(a) Train/Valid $T_0$          (b) Train/Valid $T_1$          (c) Final Test data

Figure 3: Illustration of 2 tasks out of 10 of the Spurious CIFAR2 scenario. A new environment has a new colour of spurious correlation and a new position of the spurious feature.

the percentage of images with the spurious feature to control the correlation between the spurious feature and the labels.

**Note:** A correlation of 1 means that all images have a coloured square. Hence, the probability $r(\mathcal{D}, z, y) = P(Y = y | w(z, x) = 1; x \sim \mathcal{D}) = 1$ but the probability $P(w(z, x) = 1 | Y = y; x \sim \mathcal{D})$ vary. The features created are always spuriously predictive of the labels but on various percentages of samples per class. $P(w(z, x) = 1 | Y = y; x \sim \mathcal{D})$ as the "spurious correlation" between the spurious feature $z$ and the labels $y$.

**Scenario:** The scenario is a sequence of SpuriousCIFAR2 datasets with different colours of spurious features. The test set of each scenario is the binarized version of CIFAR10 test set without any spurious features. We created the sequence of tasks using the *Continuum* library (Douillard & Lesort, 2021). We illustrate two environments and the test set in Fig. 3, and the data generation process in Fig. 2. The scenario is close to a domain-incremental scenario (van de Ven & Tolias, 2019), since the class space does not change from one task to another. However, in opposition to usual domain incremental

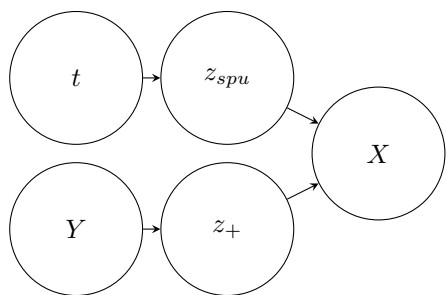

Figure 2: Data generation model of SpuriousCIFAR2 scenario. A good feature $z_+$ is generated from the labels and $z_{spu}$ is generated from the task label.

scenarios, the evaluation is not realized with one test set per task but with a common out-of-distribution test set. The evaluation is realized to track better if the out-of-distribution generalization of the model improves through time and tasks.

## 4.2 Approaches

First, we experiment with a classical vanilla replay (rehearsal) method and simple finetuning. The finetuning baseline consists of training with Adam optimizer without any CL mechanism. For rehearsal, the replay buffer is constructed by randomly selecting $N$ samples per class. The buffer is then sampled to keep class distribution uniform over all classes seen so far. Balancing sample distribution over classes is made to avoid the challenge of training on imbalanced datasets (cf Sec. G in the appendix for details).

For the existing OOD approaches, we compare continual versions of IRM (Arjovsky et al., 2019) and the state-of-the-art OOD classification methods IB-ERM, IB-IRM (Ahuja et al., 2021), GroupDRO (Sagawa et al., 2019) and Spectral Decoupling (Pezeshki et al., 2020). OOD approaches are algorithms designed to be trained on multiple environments in a multi-task fashion. IRM (Arjovsky et al., 2019) (invariant risk minimization) is an approach that uses multiple environments to learn invariant features and improves empiric risk minimization (ERM) generalization. IB-ERM augments this approach with a regularization term

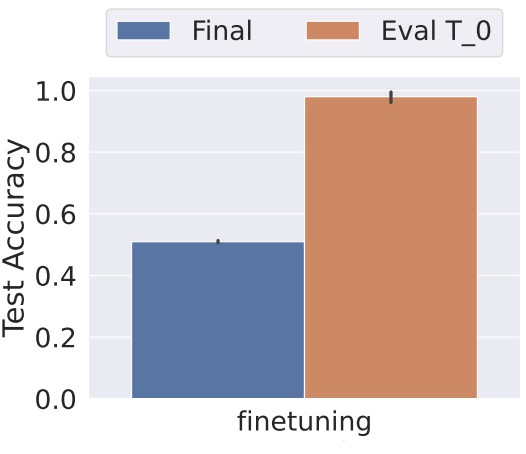 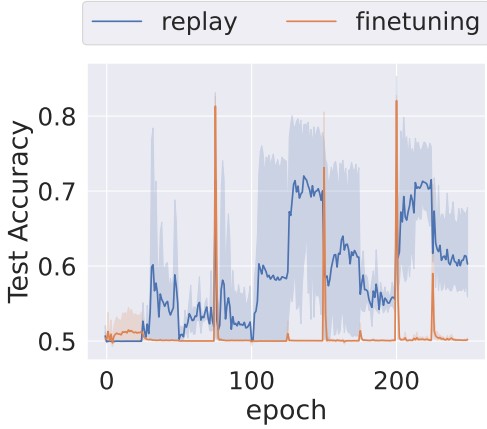

(a) Overfitting the spurious features. Performance on held on set with spurious features (Eval T_0) and without spurious features (Final) on a single task setting.

(b) Generalization and forgetting: test accuracy in a sequence of tasks (25 epochs per task) with data replay and with simple finetuning in the presence of spurious features in train data.

Figure 4: SpuriousCIFAR2 experiments: overfitting, generalization and forgetting of deep neural networks.

based on an **i**nformation **b**ottleneck constraint. GroupDRO (group Distributionally Robust Optimization) learns models that minimize the worst-case training loss over the set of environments. (Sagawa et al., 2019) proposes to couple group DRO models with a strong regularization term, such as L2 regularization or early stopping, to improve generalization. Spectral Decoupling (Pezeshki et al., 2020) proposes a regularization term that maximizes the number of features an algorithm learns to avoid relying only on spurious features. The continual version of OOD approaches simulates multiple environments by replaying data of past tasks. The adaptation of all those methods is then to add a replay buffer to the algorithms to train the baseline continually through the sequence of tasks. The replay buffer simulates the growing number of environments for the OOD approaches. In such a context, replay (rehearsal) is equivalent to the ERM (empiric risk minimization) baseline in OOD literature. We empirically choose a replay buffer storing 100 samples per class for both OOD approaches and vanilla replay (cf appendix D for HPs selection).

### 4.3 Experiments

#### 4.3.1 Problem Highlights

**Overfitting the spurious features:** To assess that algorithms overfit on the spurious features, we train the model on a single task with spurious features. We compare the test accuracy on data without spurious features (final test set) with the test accuracy on data with spurious features (evaluation set). If the test accuracy is good with spurious features and bad without, the algorithm overfits the spurious features. Fig. 4a, show exactly this phenomenon, test accuracy on task 0 is near-perfect accuracy, while on the final test set, the accuracy is near-random prediction. This figure shows that the artificial spurious features cause the expected learning behaviors: the model overfits and generalizes poorly.

**Instability:** In Fig. 4b, we assess the test accuracy $A$ at each epoch over the whole sequence of 10 tasks. This figure indicates two interesting pieces of information. First, even in the 100% spurious correlation, i.e. all images have a square of colours, baseline models can learn at some point a good solution. Secondly, even when they learn a good solution, they are very unstable and can easily forget a good solution. To lighten the influence of instability on the evaluation metrics, in the later experiments of this section, we report the average test accuracy after each task, which we note $\Omega = \frac{1}{T} \sum_{i=0}^{T-1} A_i$ instead of reporting the final test accuracy $A_{T-1}$.

**Comparing CL Baselines with OOD Baselines:** We now assess how baselines adapted from the OOD field behave in the 100% spurious correlation setting with 10 tasks. In our experiments, *IB-ERM*,

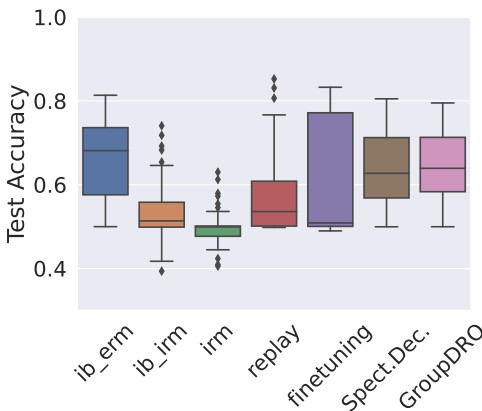

Figure 5: Accuracy with local 100% correlation between spurious features and labels

*SpectralDecoupling* and *GroupDRo* show some interesting improvement over rehearsal and finetuning baselines ( Fig. 5). On the other hand, *IRM* and *IB-IRM* performed poorly.

Those experiments show us that we can have some improvement over replay and the finetuning baseline when there is 100% correlation between spurious features and labels. Note that there is much variance in results because of the instability mentioned earlier. Moreover, the improvements stay far from a satisfying accuracy. Indeed, the average performance of the baselines on SpuriousCIFAR2 is below 70% of average accuracy. Meanwhile, we trained a model on the CIFAR2 dataset without spurious features and reached 96.73% of accuracy. The next experiments will analyze performance while gradually growing the spurious correlations.

### 4.3.2  Influence of Spurious Correlation

In this section, we aim to answer the question "how does the level of spurious correlation influence learning algorithms?". From a stream of tasks, we can expect that the 100% spurious correlation is quite rare. Hence, we will investigate the setting of lower spurious correlation in this set of experiments. We study the 25%, 50% and 75% spurious correlation cases along the 100% case.

Fig. 6 show that lowering the spurious correlation makes rehearsal and finetuning the best approaches. Those results indicate that the OOD baselines are most interesting in the very high spurious correlation setting but are not very interesting when the spurious correlation is lower or equal to 75%. It might seem counter-intuitive that finetuning is over the best baseline in a continual learning setting. Still, only the spurious correlation change in our scenario, so the global features needed to solve all tasks are in each task. It is then reasonable that with a lower spurious correlation finetuning works well. One interesting result from those experiments is that beyond 100% of correlation, varying the share of samples with spurious features has a low impact on performance. Indeed, as shown in Fig.6, experiments with 25%, 50% and 75% spurious correlation have very similar results. This lead to the conclusion that spurious features arising on less than 75% of data points can be ignored.

**Note:**  The scenario could be made harder by only keeping a subset of the dataset in each task, i.e., lowering the support of the data distribution (cf in appendix B). Indeed, lowering the support makes the task harder because there are less data per task and removes the full observability of data. Good/global features are no longer always observable, and some of them can be observable in some tasks but not others.  It is similar to creating several environments with separate splits of the full distribution as in DomainBed datasets (Gulrajani & Lopez-Paz, 2020).

### 4.3.3  Potential Solutions to Lower Impact of Spurious Features

A solution to prevent models from relying too much on spurious features is (1) to force them to learn/use more features or (2) to try to ignore the spurious features. We experiment with (1) by using a regularization

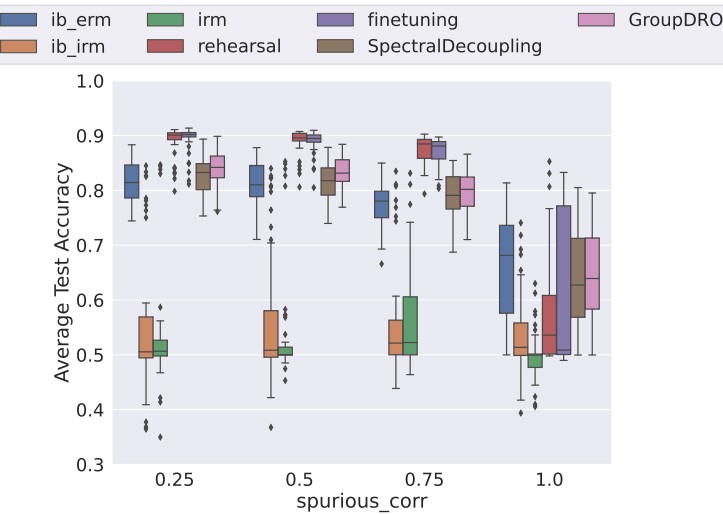

Figure 6: Averaged accuracy $\Omega$ on 10 tasks over various amount of spurious correlation between spurious features and labels.

strategy that maximizes the number of features selected. This regularization is different from regularization methods designed to not forget. We experiment with (2) by using a pre-trained model on a trusted task. The idea of the pre-trained model is that it will ignore noisy features, such as SFs.

**Using a pre-trained model:** We can use a pre-trained model on a trusted data source to ignore spurious features. For example, the spurious feature in our setting is noise, so if we use a pre-trained model on a known dataset such as CIFAR100, the model filter out noisy features and can significantly improve results. This approach, experimented in Fig. 7a, shows clearly that using a pre-trained model (here frozen) can erase the problem of noisy spurious features. This solution is convenient, but it assumes we have a compatible trusted set of data (or a trusted model) and that the spurious features

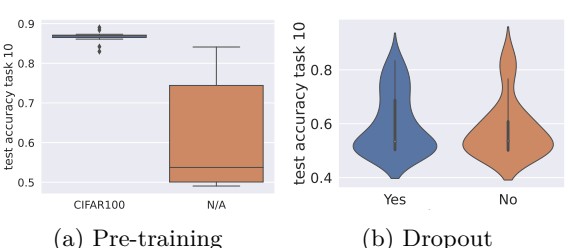

(a) Pre-training    (b) Dropout

Figure 7: Canceling noisy spurious features with pre-trained models.

are noisy. More details on noisy features are in appendix 3.4.

**Maximizing the amount of features selected:**

If we can not use the previous solution, another potential solution is to learn to select as many features as possible that could help solve the problem. A famous solution to maximize the features learned is **dropout**. Dropout randomly replaces some activations value with a zero for inference to force the model to learn more robust features. It has notably been widely experimented in continual learning (Goodfellow et al., 2013; Mirzadeh et al., 2020). We experimented with 0.25, 0.5, and 0.75 amounts of dropout just before the last linear layer. However, in our experiments, it did not show any improvement (cf. Fig. 7b).

Nevertheless, on a similar idea as dropout, the **spectral decoupling** approach is designed to address the gradient starvation problem. The gradient starvation problem arises when the loss is minimized by capturing only a subset of features relevant to the task, despite the presence of other predictive features that fail to be discovered (Pezeshki et al., 2020). Spectral decoupling is designed to discover supplementary features even with minimal train error. As dropout, it enables the possibility to learn additional features that could help to improve the test error. The experiment in section 4.3.1 illustrated in Fig. 5 indeed shows that in the 100% spurious correlation experiment, this strategy greatly improves simple rehearsal proving the potential of the idea.

**Summary:** We proposed some trivial solutions to illustrate how supplementary knowledge or assumptions on the spurious features might help prevent or fix bad learning behavior. However, it would probably not work as easily in a setting with more complex spurious features.

Experiments in sections 4.3.2 and 4.3.3 investigate how CL algorithms can deal with spurious features. We created a benchmark with spurious features and empirically investigated algorithms' performance when varying the correlation of spurious features with labels. In the next section, we will investigate local spurious features, which are a type of spurious feature specific to CL. We will investigate if those features may cause a performance decrease in CL algorithms. We show that models can overfit spurious features easily, in the next section, we investigate if local spurious features (LSF) can exist in existing original datasets and if they can easily be overfitted and influence models' performance.

## 5 Influcence of Local Spurious Features (LSF) in Continual Learning

In the previous section, we study how spurious features impact continual learning algorithms. In this section, we show that local spurious features (LSFs) may also lead to a performance decrease in usual continual learning scenarios.

### 5.1 Local Spurious Features Setting

Our goal is to investigate if some LSFs influence performance in continual learning scenarios without being manually added. To demonstrate their influence, we will create a setup where we can disentangle their influence from other factors. LSFs should lead a model to learn correct solutions to tasks independently, but those solutions can not generalize to tasks together.

**Data:** We experiment with classical datasets without modification to create class-incremental scenarios as in most continual learning literature. We use CIFAR10, OxfordPet, OxfordFlowers and CUB200 datasets. We create scenarios by splitting each dataset into 5 tasks with disjoint sets of classes.

**Model:** We use frozen pre-trained models: a resnet model on CIFAR100 for CIFAR10 and for the other datasets, we use VGG, Resnet, Alexnet and googlenet pre-trained on Imagenet from torchvision library. We assume that the pre-trained models provide a feature space sufficient for the continual downstream scenarios. They are used frozen with a linear classifier on top. We also assume that algorithms select features that are statistically predictive of labels to solve the current task. **The goal is then to show that classifiers rely on local spurious features while learning continually.** If it is the case, this will lead to a good performance on the tasks but not in the full scenario. The training is realized with simple finetuning without any replay or regularization.

**Approach:** The training is realized in a multi-head fashion with one independant linear classifier per task. Class probabilities are estimated by applying the softmax function to each head independently. We note the test performance in *multi-head* $A_{te,L}$. It estimates the capabilities of the features selected to solve tasks independently and locally.

After each task, we freeze the weights of past heads to avoid forgetting. In the end, for final evaluation, we merge all the heads into a single linear classifier and evaluate this classifier on the whole scenario (the test performance with a single head is noted $A_{te,G}$). In this second evaluation, the softmax function is applied to all output of all heads at once. This second evaluation estimates the generalization capabilities of the features selected in each head.

Hence, if classifiers select local spurious features, $A_{te,L}$ should be significantly bigger than $A_{te,G}$.

### 5.2 Disentangling Factors of Influence

Apart from the problem of overfitting LSFs, a gap between $A_{te,L}$ and $A_{te,G}$ could be explained by (1) the difference in the difficulty of both evaluations (multi-head and single head), (2) an unbalance of bias and norms from different heads that could lead to bad performance in *single head* (more details about this problem

in appendix E). However, if neither (1) nor (2) are sufficient to explain the performance gap, then we can conclude that LSFs influence the performance of the model.

**(1) Difficulty gap:** the comparison between single-head and multi-head is biased because the first is one 10-way classification (harder) while the latter is the addition of five binary classifications (simpler). To estimate the difference of difficulty between both, we added a non-parametric method: a nearest mean classifier, denoted here *MeanLayer*. There is no feature selection in MeanLayer, as the classifier only uses the mean of the features of each class. Consequently, the difference in performance with MeanLayer in multi-head and single-head depends mostly on the difference in the difficulty of both evaluations. We use the meanlayer accuracy as a proxy for the evaluation gap.

**(2) Unbalance of bias and norms:** we compare the linear layer performance with the weightnorm layer from (Lesort et al., 2021). This layer does not use norm and bias for inference and is then insensitive to such imbalance (details in appendix E).

We note that forgetting is, by design, impossible here since the features extractor and the other heads are frozen. Hence, forgetting can not explain drops in performance.

### 5.3 Local Spurious Features Experiments

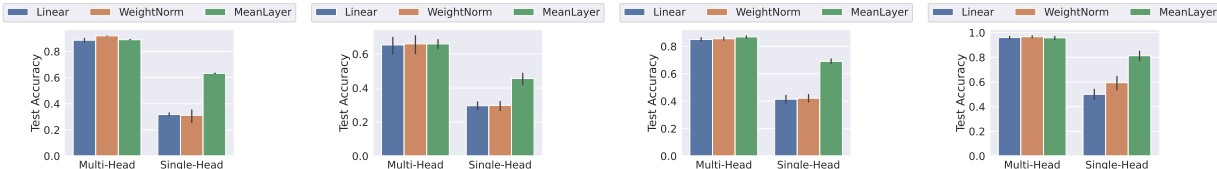

Figure 8: Comparison of final test accuracy $A_{T-1}$ in multi-head and the same classifier without task label information (In this order: CIFAR10, CUB200, OxfordFlower102, OxfordPet). The results are averaged among several pre-trained models. The performance gap in the linear layer indicates how the local features selected are generalizable. We use the weightnorm to estimate if bias or norm imbalance phenomena play a role in the performance gap. The MeanLayer results is a proxy to estimate the difference in the difficulty of both evaluations.

We report results in Figure 8. We see that there is a significant accuracy gap between multi-head (MH) $A_{te,L}$ and single-head (SH) $A_{te,G}$ performance in the linear layer.

On the one hand, the gap of performance is similar for the linear layer and the weightnorm layer. On the other hand, the gap for MeanLayer is significantly smaller than the gap for the linear layer. Hence, the factor of influence (1) and (2) from the previous section can not explain the accuracy gap of the linear layer. We conclude that the local spurious features are responsible for this gap. The model overfits some of them, leading to a poor generalization of solutions learned locally.

These experiments prove that the drop in performance in the usual class-incremental scenarios can be due to a bad presence of local spurious features and not necessarily to forgetting. This is a fundamental observation for future continual learning approaches as it means that the features learned in one task should not necessarily be preserved for later tasks.

Another interesting insight that this experiment gives is that contrary to experiments in section 4.3.3, which show that using a *pre-trained model can help for spurious features, it does not offer a solution to local spurious features.* A good feature extractor is then not sufficient for a good feature selection. The details of all the results are in appendix F.

### 5.4 Effect of Number of Classes per Task

In previous experiments, we have showed that spurious features influence performance. In this section, we assess if this problem is vary tasks become harder. We propose to increase the number of classes per task to increase complexity. The hypothesis is that the model will be forced to learn more about the data and

learn more robust features to create a more complete representation of the data decreasing the influence of local spurious features. We create scenarios of 5 tasks with various numbers of classes per task from tiny-Imagenet and CUB200. We train only a classifier on top of a pre-trained model in a similar way as in previous experiments (Sec. 5.1).

Fig. 9 reports the difference $\Delta_{Head} = A_{te,L} - A_{te,G}$ for a various number of classes per task. For each scenario, one task represents 20% or the total number of classes. Hence, the difference in difficulty between solving tasks separately (MH) and together (SH) is assumed to be marginal. The results with MeanLayer in Fig. 9 confirm this hypothesis; the gap stays quite stable with the number of classes per task. On the other hand, Fig. 9 shows that increasing the number of classes per task reduces $\Delta_{Head}$, which means that the influence of local spurious features is reduced. In practice, the algorithm designers should be more careful about local spurious features when the number of classes in a task is low.

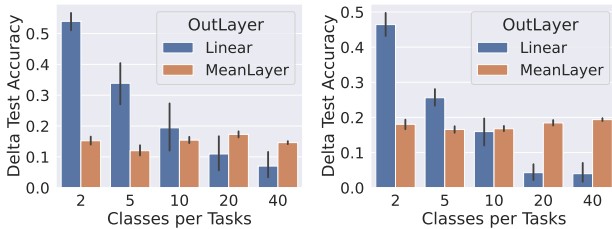

Figure 9: Delta accuracy between MH and SH accuracy. The effect of LSF reduces with the number of classes per task. CUB200 (left), TinyImagenet (right), 5 tasks per scenario.

**Summary:** In this section, we have shown that local spurious features (LSF) influence training in usual scenarios of continual learning and lead to performance decrease. We proved it by highlighting their influence on a classifier on top of frozen pre-trained models. If, with a pre-trained model, this problem arises, we can assume that the same problems would also exist when training models end-to-end continually. We also showed that the influence of LSF depends on the number of classes inside a task. The more classes there are inside a task, the less influence LSFs have.

## 6 Discussion

**Spurious Features vs Local Spurious Features:** Spurious features and local spurious features lead to the same problem for learning algorithms: overfitting features that are not discriminative. The difference is that spurious features result from a covariate shift between training and test data. In contrast, local spurious features are due to the unavailability of all data while learning. Local spurious features are then, more specifically, a continual learning challenge.

**Solutions to local spurious features:** The problems of local spurious features lead to phenomena where forgetting is helpful to improve final performance (Zhou et al., 2022), also denoted as graceful forgetting. Indeed, forgetting by reinitializing some weights allows escaping spurious local minima and relearning a better solution taking new data and knowledge into account. On a more general note, local spurious features make ineffective approaches that are too rigid and unable to modify and fix previously learned features/knowledge. We recommend using replay as a practical solution to reduce SPs and LSPs influence. Indeed, the replay process can simulate an identically and independent distribution of tasks seen so far. It makes it possible to fix potential mistakes in past feature selections. Nevertheless, even if this solution already exists, it significantly increases the computational cost when the number of tasks grows and becomes inefficient if SPs are too correlated with labels. Therefore, future work should aim at optimizing replay to identify and fix spurious features efficiently.

**Benchmark:** The scenario SpuriousCIFAR2 proposed in this paper has been designed to highlight the problem that spurious correlation might create in continual learning. This scenario plainly fulfils its task of disturbing CL algorithms, particularly in the 100% correlation setting. However, it can not be used as a benchmark to evaluate the robustness of algorithms. The spurious features are very simple, and a simple ad hoc processing of data could solve this scenario, i.e., encoding data with a pre-trained model as in Fig. 7a. A proper benchmark to assess robustness to spurious correlation would propose spurious features that are easy to learn by the model but harder to detect or ignore than simple squares of colour. DomainBed datasets

(Gulrajani & Lopez-Paz, 2020) are an interesting set of benchmarks for spurious correlation investigation. However, the amount of features is not controllable, making it harder to evaluate the limits of algorithms.

# 7 Conclusion

Continual learning algorithms are designed to learn and store knowledge for later use. In classification tasks, these algorithms sequentially learn to solve problems by identifying predictive features and memorizing them. However, memorizing bad features can harm future performance.

One type of bad features are the spurious features (SPs), which are highly predictive in training data but not in testing data, making models prone to overfitting. In continual learning, another type of bad feature can emerge: the local spurious feature (LSP), which is only predictive of labels within a subset of data.

Our experiments in domain incremental and class-incremental setups demonstrate that the presence of spurious features and local spurious features significantly impact continual learning algorithms. While we artificially added the spurious features in our experiments, we found out some LSPs are already present in usual class-incremental scenarios and influence results. Catastrophic forgetting is often pointed out as the main cause of performance decrease in continual learning. Our results suggest that SP and LSPs may also play a major role. Understanding this phenomenon is crucial for improving memorization mechanisms and avoiding bad features.

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

## A  Compute

The experiments were run on internal cluster on Quadro RTX 8000 GPUs. The total time of compute for an approximate period of 100 days of GPU use with hyper-parameters selections and experiments.

## B  Lowering the Support of the distribution at each task

### B.0.1  Lowering the Support Amount

In experiments Sec. 4.3.2, the same data support was used for all tasks. This means that the shared support of original data is 100%. However, we expect continual algorithms to learn with less support. The support of a task is the percentage of the full original distribution within the task. We investigate in those experiments the influence that the support amount has on learning algorithms. We set the spurious correlation to 75% in those experiments.

As it is designed in the SpuriousCIFAR2 scenario, each class (0 and 1) has 5 distributional modes corresponding to the 5 original classes. In the previous experiments, all the data for all those modes are available in all tasks only the spurious features changed from one task to another.

In those experiments, we reduce the support by subsampling the original data. We select the data of a subset of the CIFAR10 classes for each task. For example, we select only cars for class 0 and only deers for class 1,

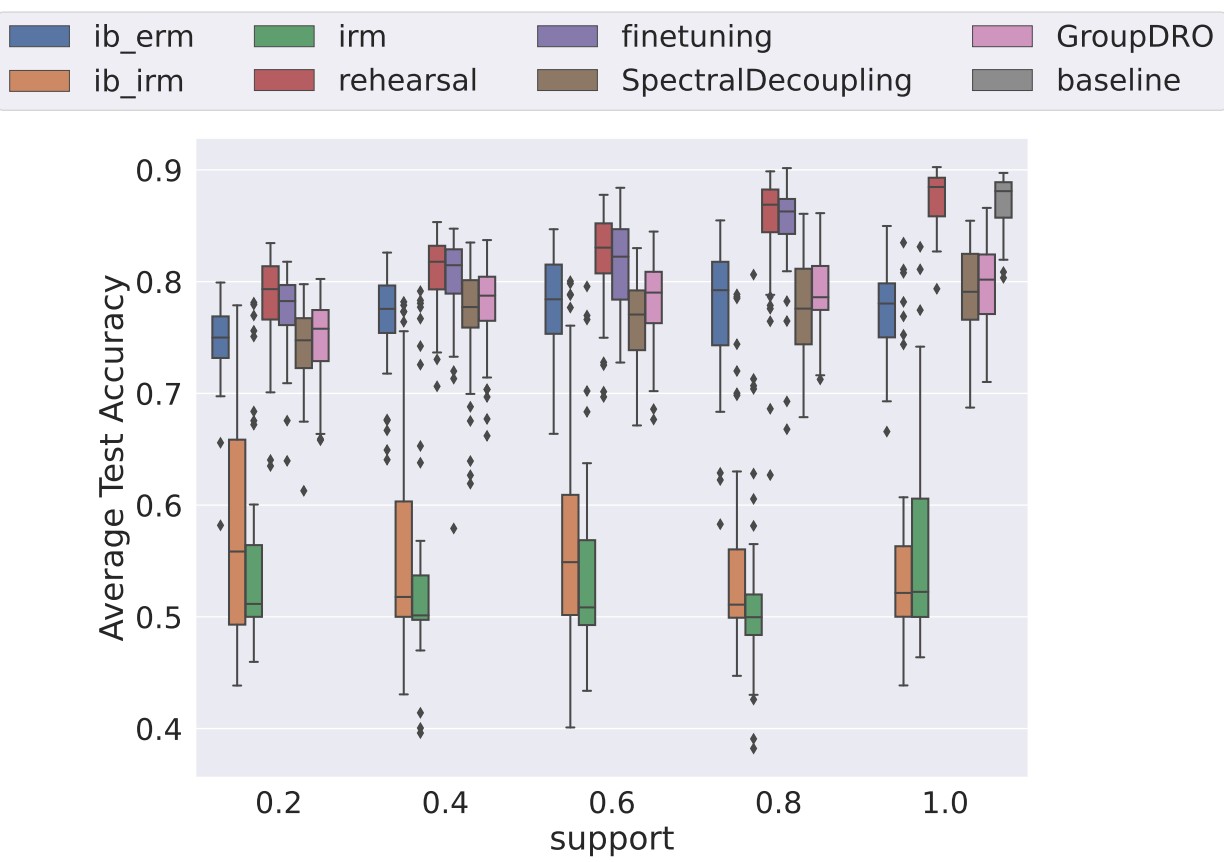

Figure 10: Averaged accuracy $\Omega$ on 10 tasks over various amounts of support, the spurious correlation is set to 0.75.

instead of airplanes, cars, trucks, ships, and horses for class 0 and birds, cats, dogs, deer, and frogs for class 1. Hence, if we use the support of 0.2 (i.e. 20%), we will only select the data of $5 * 0.2 = 1.0$ original classes for class 0 and one other for class 1. For simplicity's sake, we will use only support compatible with the number of classes to have a round subset, i.e., $[0.2, 0.4, 0.6, 0.8, 1.0]$. At each task, the support is randomly sampled, hence the same original data can be in several tasks, but the spurious features will still be different for all tasks.

The support experiments results, illustrated in Fig. 10, show that in all support settings, the finetuning baseline and rehearsal are the best-performing methods. On the other hand, contrary to what would be expected, the amount of support seems to not play an important role in the final performance, at least in the range of support possible in our scenario. This is probably because doing replay converts a partially observable setting into a fully observable setting by simulating an iid distribution.

## C  Samples Support Experiments

## D  Hyper-Parameters selection

For a fair comparison between algorithms originally designed for continual learning, such as replay, and OOD algorithms, we conduct the hyper-parameter search more intensively for OOD approaches.

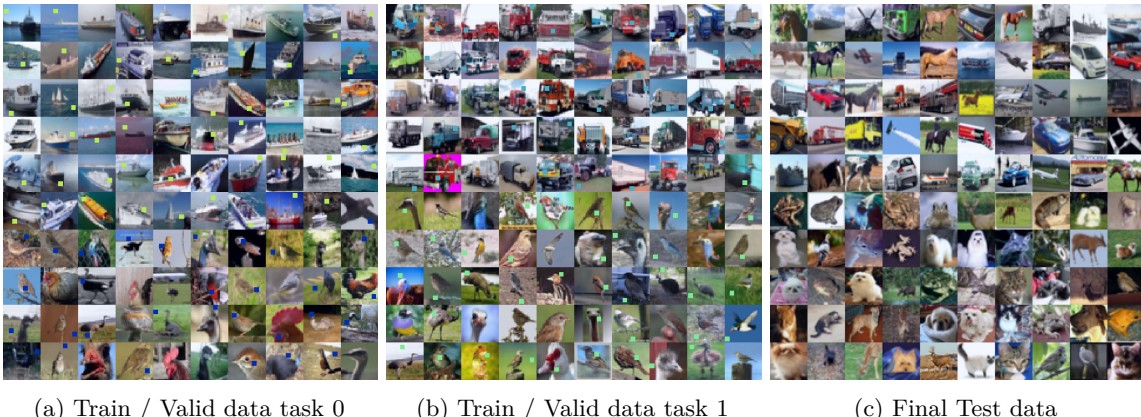

(a) Train / Valid data task 0    (b) Train / Valid data task 1    (c) Final Test data

Figure 11: Samples for support experiences, here with 20% support, i.e. the data of only two of the original classes in each task.

For each OOD approach, we search for the best hyper-parameters in the range proposed in the DomainBed GitHub repository [1]. We also searched for the best learning rate with the bayesian method of wandb (Biewald, 2020). We used approximately 100 runs for each OOD baseline to select hyper-parameters. The scenario for hyper-parameters selection was an OOD setting with 5 environments of SpuriousCIFAR2 with 75% correlation but all simultaneously available (with no continual stream of tasks).

The hyper-parameters for rehearsal and finetuning, have been selected on a finetuning training in a single task setting with 75% of correlation. The number of samples per class for replay has been selected on a 5-task SpuriousCifar2 scenario with 75% of correlation. The optimizer used was Adam with a learning rate of 0.06 (other HPs are default ones in pytorch library).

## E    Bias Norm imbalance

As described in (Lesort et al., 2021): A linear layer is parameterized by a weight matrix $A$ and bias vector $b$, respectively of size $N \times h$ and $N$, where $h$ is the size of the latent vector (the activations of the penultimate layer) and $N$ is the number of classes. For $z$ a latent vector, the output layer computes the operation $o = Az + b$. We can formulate this operation for a single class $i$ with $\langle z, A_i \rangle + b_i = o_i$, where $\langle \cdot \rangle$ is the euclidean scalar product, $A_i$ is the $i$th row of the weight matrix viewed as a vector and $b_i$ is the corresponding scalar bias.

It can be rewritten:

$$\|z\|\|A_i\| \cdot cos(\angle(z, A_i)) + b_i = o_i \tag{3}$$

Where $\angle(\cdot, \cdot)$ is the angle between two vectors and $\|\cdot\|$ denotes here the euclidean norm of a vector.

Then, at inference time, $y_i = argmax_i(o_i)$ rely on the norm of $\|A_i\|$ and on the bias $b_i$. Within a single task, i.e. within a single head in a multi-head setting, $\|A_i\|$ and $b_i$ are balanced to predict class correctly. However, we can not ensure that $\|A_i\|$ and $b_i$ will are not biased from one head to another.

To avoid unbalance for bias and norm for inference, (Lesort et al., 2021) proposed the *weigthnorm* layer where: $\|z\| \cdot cos(\angle(z_t, A_i)) = o_i$ and show that this layer is efficient in learning in incremental and lifelong settings.

---

[1] https://github.com/facebookresearch/DomainBed

---

**Algorithm 1 Balanced Sampling of Data Mixture.**

---

1: **procedure** GET_SAMPLER($\mathcal{D}$)
2:      $y \leftarrow \mathcal{D}.y$                 ▷ Get data class labels
3:      nb_per_class = bincount(y)           ▷ count the number of occurence of each class
4:      $weights\_per\_class = \frac{1}{nb\_per\_class}$
5:      sample_weights = weights_per_class[y]      ▷ give a sample probability weight to each data point
6:      sampler = Sampler($\mathcal{D}$, sample_weights, replacement=True)    ▷ create sampler to sample accordingly to the
    sample_weights
7:      **return** sampler
8: **end procedure**

---

# F    Details Multi-Head experiments

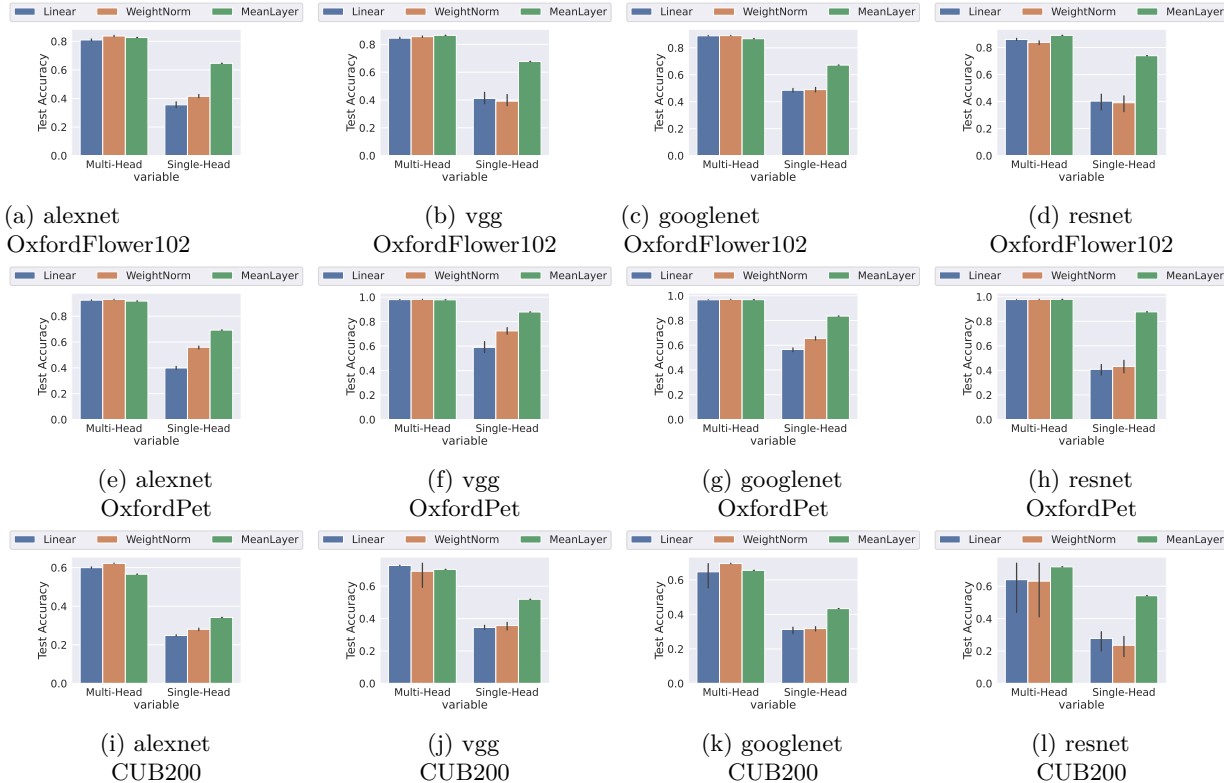

(a) alexnet
OxfordFlower102

(b) vgg
OxfordFlower102

(c) googlenet
OxfordFlower102

(d) resnet
OxfordFlower102

(e) alexnet
OxfordPet

(f) vgg
OxfordPet

(g) googlenet
OxfordPet

(h) resnet
OxfordPet

(i) alexnet
CUB200

(j) vgg
CUB200

(k) googlenet
CUB200

(l) resnet
CUB200

Figure 12: Local spurious features experiments: after training in a multi-head way, we compare accuracy between multi-head (soft-max applied on a subset of classes' outputs) and single head (softmax applied on all classes' outputs). The performance differences assess how the model selected local spurious features to solve tasks. Two baselines are added, (1) meanlayer, which assesses the difference in the difficulty of the two evaluations, and (2) weightnorm, which assesses if the performance difference is due to an imbalance in norm or bias. The results in this figure show that models indeed rely on local spurious features to solve tasks.

# G    Sampling Algorithm

Algorithm 1 describes the sampler used to train the model with replay. The input dataset $\mathcal{D}$ is a concatenation of the buffer with the current data. The goal is to make the probability of sampling on each class, the same whatever the number of samples for each class in $\mathcal{D}$.

