# OpenReview forum: "Spurious Features in Continual Learning"
_TMLR — Rejected by TMLR_

### Review · Reviewer_ciP2 · 2023-02-02

**Summary Of Contributions:**

(1) A new dataset, SpuriousCIFAR2, created to demonstrate spurious features.

(2) Modify several OOD algorithms for the continual learning setting with replay/rehearsal.

(3) Claim that local spurious features are an issue in continual learning.

**Audience:**

Yes

**Claims And Evidence:**

No

**Requested Changes:**

(1) Major revision is needed to polish the writing.

(2) Explain the motivation of CIFAR2 and clarify the experiment results/details.

(3) Further justification for the existence of local spurious features.

**Strengths And Weaknesses:**

**Strength**: Study an important problem and investigate several baseline approaches in the continual learning setting and identify a potential issue of local spurious features.

**Weaknesses**

(1) Writing can be greatly improved.

- The definition of z is ambiguous in the paper. Sometimes it is referred to as a single feature (e.g., Table 1 caption), while other times it is referred to as a set of features (e.g., Sec.3.3, the $z_+$ is denoted as a set of "good" features).
- Fig.1(c) seems to contradict the definition of local spurious features in Table 1: red and yellow can be used for $D_{te}$
- Sec.3.4 uses additional terms without properly defining them. What are "global features"? The notations of invariance and noisiness are introduced without proper motivations. Why do we care about them and how they can naturally appear in a learning scenario?
- Fig.4 the captions of the two subfigures are too close to each other
- Fig.6 baseline should be renamed to finetuning since "baseline" is a very generic term here.
- Sec.4.3.2, "spurious features arising momentarily on less than 75% should not perturb most approaches" rewrite needed.
- Sec.5.1: "a resnet model on CIFAR100 for CIFAR10" but CIFAR100 is not used. "Hence, while training, inference considers one head at a time for classification"? In addition, $A_{te-local\textunderscore softmax}$ is unnecessarily long (something like $A_{te,l}$, $A_{te,g}$ would suffice)
- The captions of Fig.7 and Fig.9 almost overlap with the main text

(2) The motivation for converting CIFAR to a binary problem is unclear.

- In Sec.4, why CIFAR2 when we can do 10? What’s the main purpose of reducing the problem into a binary classification problem?

(3) Some experiments need further details/results.

- Fig.6, it would make more sense to have a more-fine grained spurious correlation toward the high end (90%-100%)
- Sec.4.3.3: the first paragraph suggests forcing them to learn/use more features based on regularization. However, the term regularization usually would reduce the number of features in the literature.
- Sec.4.3.3: is the pre-trained model used as the initialization for sequential training (i.e., the whole model is unfrozen)?

(4) The discussion on local spurious features in Sec.5.1 is questionable. The results in Fig.8 can also be explained by the (non-local) spurious features. For example, FeatureA is used for Task1, FeatureB is used for Task2 and when combined, either of them is sufficient and we need FeatureC for the joint task. Both FeatureA and FeatureB are then spurious features rather than local spurious features. Since the model is essentially linear (a linear head on top of pre-trained and frozen features), why not just look at the coefficients of the linear model?

(5) No solution is provided for the LSF issue.

Minor typos

- Page 2 first paragraph, contribution (4): along with CF -> along with SF?
- Sec.5.1: CUB200 datasetsWe -> CUB200 datasets. We…
- There are three dots in each subfigure of Fig.8, which if copied, read “variable”

---

### Review · Reviewer_prpm · 2023-02-15

**Summary Of Contributions:**

This paper considers a local spurious features in the context of continual learning. The major contributions that the authors claimed in the paper include:

(1) introduce the notion of local spurious features in continual learning, which differs from the general spurisous features caused by the data covariate shift.

(2) create a new variant dataset based on CIFAR-10 for experimentation.

(3) improve a OOD generalization method for continual learning and conduct its experiments.

(4) identify local spurious features as a core challenge for continual learning.

**Audience:**

Yes

**Claims And Evidence:**

No

**Requested Changes:**

- I think the experiment section can be further improved. For example, some attention-based visualization (say Grad-Cam) can be used to verify the network indeed overfits to the added color block. Only test accuracies are reported, and they convey too little information for one to draw any conclusion.

- The presentation of experiments is confusing and messy. I suggest to re-organize the experiment, polish the figures/tables and state the conclusion more clearly.

- For the other requested changes, see the weakness part.

**Strengths And Weaknesses:**

First of all, continual learning is an important problem and serves as a stepping stone to go from i.i.d. learning to non-i.i.d. learning. This paper considers a concept of local spurious features, which is explained intuitively in Figure 1.

**Strengths**:

- The paper is well written and easy to follow. Explanations of the core idea is clear.
- The idea of tackling continual learning from the perspective of spurious features sounds interesting. And I do agree continual learning has close connection to train/test data covariate shift.

**Weaknesses**:
- Despite the interesting idea, I fail to understand why understanding continual learning from a spurious feature view will be benefiticial. Does it provide any new insights on how we can improve continual learning algorithms? It doesn't seem to provide many new insights at this stage.
- I have concerns on the necessity of introducing the notion of local spurious features. From my perspective, local spurious features are the same as spurious features, in the sense that both of them are spurious correlation in terms of tasks (rather than data/class). The only difference is that normal spurious features are for train/test distribution shift or OOD generalization (viewed as a single task), while local spuirous features are for sequential tasks in continual learning. There is no crucial difference here.
- Claims in the experiment section are too vague. For exmaple, the solution mentioned in 4.3.3. It lacks convincing empirical support, and moreover, it suggests no concrete solution.

---

### Review · Reviewer_5kXM · 2023-02-15

**Summary Of Contributions:**

This paper presents an empirical study of the so-called spurious features in the context of continual learning. In particular, the authors discuss spurious features (SF) and local spurious features (LSF), which cause poor generalization performance in continual learning. In the experiments, the authors design corresponding experiments to verify the impact of SF and LSF respectively and conclude that both SF and LSF result in performance drop in continual learning.

**Audience:**

Yes

**Claims And Evidence:**

No

**Requested Changes:**

Please follow the weaknesses part. Although the topic is interesting, from the current experimental design, I cannot really tell if spurious feature really exists/matters in the continual learning setting.

**Strengths And Weaknesses:**

Strengths:
- The topic of studying spurious features in continual learning is interesting.

Weakness:
Since the paper is mainly about empirical investigation, I expect the experimental results being clear persuasive. However, existing results are not quite solid and some seems a bit confusing.
- The design of "SpuriousCIFAR2" is questionable.
  - As a continual learning dataset, it contains only 2 tasks (in a domain-incremental way), which makes it not that different from the usual supervised learning setting.
  - Why making it a binary classification problem? It seems the original 10-way classification task is compatible with the add-on color squares.
  - Adding color square is merely valid for creating a synthetic dataset. However, in real world settings, we will never encounter the setting. I recommend creating a more realistic dataset from, for example, DomainNet.

- The study of local spurious features (LSF) is very confusing.
  - Most importantly, the conclusion "we have shown that local spurious features (LSF) exist" is not solid. The complex comparison design and tricks to disentangling factors of influence (which are probably not exhaustive) make the conclusion weak. The authors should **directly** show the existence LSF, for example, by explicitly selecting and visualizing the spurious features for every task.
  - The experimental design only focus on the linear classifier (either single or multi-head). What about nonparametric methods like prototype based classifiers?

- Since there are so many different settings in continual learning, please make the base setting clear. For example, Sections 4 is more like domain-incremental while Section 5 bases on task and class-incremental (single/multi-head). Include the settings and assumptions explicitly into the conclusion can make the scope more clear.

- The writing and formatting should be improved, for example:
  - The axis label fonts of Figure 5 and 6 should be unified.
  - Subsection title of 5.2 should be fully capitalized.
  - Space below Figure 2 is too much, while the contrary for Figure 9.
  - Notations like $A_{te-local/global-softmax}$ should be much more concise.

---

### Decision · Action_Editors · 2023-03-22

**Recommendation:** Reject

**Comment:**

While all three reviewers agree that this is an important topic, they raised the following problems:
1) the experiments are not very well motivated, and the experimental setting is not always completely clear [prpm, 5kXM and ciP2].
2) some conclusions drawn from the experiments on the role of LSF are actually questionable [prpm, 5kXM and ciP2].
3) finally, the relevance of LSF itself is questioned: is seems quite similar to SF [prpm], moreover, no specific solution for LSF is proposed [ciP2].

We all agree that there might be an interesting notion here, but that evidence to support the relevance of LSF is missing. All reviewers agree that a much stronger set of experiments would be necessary for this. Thus, the paper cannot be accepted for publication in TMLR.

The reviewers provided constructive comments that can be used to improve this work. I recommend to take these comments into account before submitting the paper to another review. I wish good luck to the authors for this new submission.

**Audience:**

Large community working on lifelong learning and catastrophic forgetting.

**Claims And Evidence:**

The papers study the problem of catastrophic forgetting in lifelong learning. The role of spurious features (SF) in catastrophic forgetting was highlighted in previous works. In addition to the usual notion of (global) spurious features (SF), the authors introduce a new notion of local spurious features (LSF). Experiments show that both SF and LSF lead to forgetting.